# Effects of Motivational Climate on Knowledge Hiding: The Mediating Role of Work Alienation

**DOI:** 10.3390/bs12030081

**Published:** 2022-03-16

**Authors:** Soojin Lee, Xinzhu Yang, Jinhee Kim, Gukdo Byun

**Affiliations:** 1College of Business Administration, Chonnam National University, Gwangju 61186, Korea; soojinlee@jnu.ac.kr (S.L.); youngxinzhu@naver.com (X.Y.); 196842@jnu.ac.kr (J.K.); 2School of Business, Chungbuk National University, Cheongju 28644, Korea

**Keywords:** knowledge hiding, motivational climate, work alienation

## Abstract

Although knowledge is arguably an organization’s most important resource, many organizations still practice knowledge hiding. This study explores how an organization’s motivational climate—mediated by work alienation among its members—influences knowledge hiding from the perspective of the conservation of resources (COR) theory. Specifically, we establish hypotheses that the performance and mastery climates, mediated by work alienation, have positive and negative effects on knowledge hiding, respectively. To verify these hypotheses, we conducted a survey among members of Chinese companies, through which 200 responses were collected through a two-wave panel design. The results of the analysis demonstrated that motivational climate, as an antecedent of knowledge hiding, has a significant effect on work alienation. We also found that work alienation mediated the relationship between (a) performance climate, and (b) mastery climate and knowledge hiding. Based on these findings, we discuss the research implications and limitations while suggesting directions for future studies.

## 1. Introduction

In today’s economy, knowledge is a vital resource that enables an organization to innovate and gain a competitive advantage [1,2]. Simultaneously, organization members consider knowledge an important asset, as they can use it to elevate their status and get rewards in the organization [2,3]. Consequently, despite companies striving to promote knowledge sharing, organizations still practice knowledge hiding [4,5]. Knowledge hiding—the intentional concealment of knowledge that has been requested by another coworker [6]—can be considered a strategy to gain a competitive edge and maximize benefits by individuals in an organization [2,7]. However, in addition to its potential negative effect on employees, knowledge hiding can cause an organization to incur substantial economic losses [4,8]. According to Babcock [9], the loss incurred from knowledge hiding totals USD 31.5 billion annually for Fortune 500 companies [10].

Against this backdrop, studies have made substantial efforts to identify the antecedents of knowledge hiding to reduce knowledge hiding behavior among organization members. However, previous studies mostly focused on interpersonal relationships, including distrust or social exchanges among team members, thus paying little attention to the effect of the organizational environment on knowledge hiding (e.g., Connelly et al., Lin & Huang, Tsay et al.) [6,11,12]. Accordingly, many studies have pointed out that organizations need to examine and understand the cause of knowledge hiding from various perspectives, to devise effective strategies to mitigate knowledge hiding in an organization [2,6,13]. This study considers that members are more likely to hide knowledge when they feel the organizational climate does not encourage knowledge sharing [14]. A climate in which organizations emphasize performance gives employees the organizational norms and requirements for success, which in turn determines whether they will hide or share knowledge with their coworkers [8].

The motivational climate at work represents just such a context: it refers to the shared perceptions of the criteria for the success or failure of employees, emphasized through the policies, practices, and procedures of the work environment [15,16,17,18]. The motivational climate comprises two dimensions: mastery climate and performance climate. A mastery climate emphasizes effort, sharing, and cooperation, while a performance climate emphasizes competition, comparison, and recognition from other people [2,16]. Such different climates, where opposing criteria for success are emphasized, may yield completely different results regarding knowledge hiding behavior among employees. This study aims to fill the research gap by highlighting motivational climate as an antecedent of knowledge hiding.

Moreover, as a key variable linking motivational climate and knowledge hiding, we present the mediating role of alienation. Work alienation is defined as an awareness of the discrepancy between employees’ perceptions of an objective work situation and their personal concerns, such as requirements, values, ideals, desires, or expectations [19]. Based on the COR theory [20], which posited that individuals try to acquire and maintain resources and protect themselves from loss in the work environment, we can infer the mechanism of alienation at work behind the relationship between motivational climate and knowledge hiding. More specifically, if they are persistently exposed to the constant threat of competition and comparison with their coworkers (i.e., performance climate), individuals may become alienated, which is characterized by a state of lack of resources [21]. This may drive them into a loss spiral; resources-limited individuals facing exhaustion may resort to strategies to defend their position by hiding the knowledge resources they have (e.g., Guo et al., Yao et al.) [13,22]. In contrast, in a mastery climate, organization members grow by learning and cooperating with other coworkers; individuals can use investment strategies to enrich their resources by providing their resources at the request of coworkers based on the norm of reciprocity. Individuals intentionally hide knowledge in a performance climate but are willing to share knowledge in a mastery climate; this study’s ultimate purpose is to investigate and understand the reason behind this disparity.

In summary, this study aims to contribute to the literature in three respects. First, we empirically demonstrate the effect of motivational climate on knowledge hiding. Second, our study verifies the relationship between motivational climate and work alienation to clarify that mastery and performance climates influence the work alienation of individuals in opposite directions. Third, we highlight the mediating role of work alienation as an underlying mechanism between motivational climate and knowledge hiding. As a result, this study’s significance also lies in its contribution to understanding how individuals decide to hide or share knowledge in a motivational climate mediated by the sense of alienation in the framework of the COR theory.

## 2. Hypothesis Development

### 2.1. Knowledge Hiding and Motivational Climate

Connelly et al. [6] first presented the concept of “knowledge hiding” in their study on “knowledge-sharing hostility” and “knowledge retention”. In this context, “knowledge” mainly refers to the information, perspective, and experience related to work, with an emphasis on the intention to hide. Therefore, although they might sound similar, knowledge hiding differs from a lack of knowledge sharing. There are roughly three forms of knowledge hiding: playing dumb, evasive hiding, and rationalized hiding [6]. “Playing dumb” refers to instances where the hider feigns ignorance to avoid giving the requestor information; “evasive hiding” occurs when the hider provides incorrect or partial information or delays the provision of information; and “rationalized knowledge hiding” refers to when a hider explains hiding information—for example, by stating that they are not authorized to provide the requested information, or that the information is classified.

This organizational knowledge hiding behavior can be influenced by how organization members perceive organizational contexts [14]. Particularly, the organizational environment regarding performance, such as motivational climate, may play a decisive role in hiding knowledge [5,6]. As noted above, motivational climate can be categorized into two types: mastery and performance [23]. A mastery climate refers to a situation where individuals perceive that efforts, sharing, and collaboration are valued, and learning and skills are emphasized in an organization [8,15,24]. Conversely, a performance climate emphasizes normative standards for success, with social comparison and competition at its core. In this environment, individuals should outperform their coworkers [2]. Considering that a motivational climate allows the organizational members to understand the kind of behavior that is expected and rewarded, it can significantly influence their decisions to hide or share knowledge [8].

There is evidence of the influence of these climates on individual members in the fields of sports and education. In a mastery motivational climate, athletes become more confident, as they do not need to show that they are better than their competitors and receive equal treatment [25]. In this climate, individuals make progress together by challenging themselves and encouraging each other, learning from errors, and sharing feedback for improvement [15]. Additionally, they are more likely to perceive themselves as significant in the organization, be self-determined, and maintain a higher vitality level [26]. However, in a performance climate, individuals perceive competition as something that reflects their capabilities and undervalues themselves when they fail; they are sometimes afraid of being punished for doing things wrong [23,25,27]. Individuals tend to become extremely competitive and believe they need to stand out [24]. Such a performance climate can cause insecurity, turnover, conflicts between coworkers, morale issues, and even cheating to succeed [28,29].

A recent study on this topic found that the mastery climate moderates the relationship between ethical leadership and knowledge hiding [5]. In a performance climate, where competition and social comparison are key, performing better than one’s peers is the criterion for success [2,8]. To avoid falling behind their peers, individuals can choose to hide knowledge required by their coworkers not only to gain competitive advantage but also to delay and hinder excellent performance among their coworkers [8,30]. In other words, they will try to defend their performance by intentionally holding back the knowledge requested by their coworkers. Consequently, knowledge hiding or the lack of sharing knowledge can become prevalent in a performance climate.

In contrast, in a mastery climate, an organization helps themselves and others while developing skills and contributing to knowledge enhancement in the workplace. This becomes the criterion of real success [8,31]. Consequently, individuals in a mastery climate consider their and their coworkers’ interests. Moreover, once learned by a member, the insights, and lessons are shared so that others can benefit from the knowledge [8]. In such a climate, intentional knowledge hiding is not beneficial to the hider, as they are likely to miss the opportunity to develop competence and improve the knowledge quality by sharing their knowledge with their coworkers [8]. Accordingly, in a mastery climate, knowledge hiding is likely to be perceived as a negative behavior that hinders individuals from gaining mutual benefits through knowledge exchange [8]. Based on this assumption, this study proposed the following hypotheses:

**Hypothesis** **1a** **(H1a).**
*Performance climate is positively related to knowledge hiding.*


**Hypothesis** **1b** **(H1b).**
*Mastery climate is negatively related to knowledge hiding.*


### 2.2. Motivational Climate and Work Alienation

Kanungo [32] defined work alienation as the cognitive separation from one’s job, frustration, and negativity from failing to achieve one’s objectives at work, and behavioral apathy. Work alienation can be categorized into powerlessness, meaninglessness, and social isolation [33]. “Powerlessness” refers to the lack of control over the incidents in one’s personal life and workplace tasks. For example, in an alienating work environment, employees cannot control the work process or be involved in the decision-making process. “Meaninglessness” means finding no value or significance in one’s own work. “Social isolation” refers to the sense of being psychologically isolated from the organization and one’s coworkers. In the same vein, Nair and Vohra [34] revealed that the strongest predictors of work alienation among knowledge workers are lack of meaningful work, work that does not allow for self-expression, and poor-quality work relationships.

From the viewpoint of the COR theory, a sense of control (e.g., feeling that one has control over one’s life), sense of meaningfulness (e.g., feeling that one’s life has meaning/purpose), and social support (e.g., feeling that one has support from one’s coworkers) are also important to an individual [20]. In a performance climate, it can be said that individuals are under “forced social comparison”, overwhelmed by comparison and divided by abilities [8,16,35]. In other words, a performance climate takes away the right of self-determination of individuals [16,36]. Therefore, in a perceived performance climate, organization members may feel that they are powerless or lack control over their work, so they lack the right to self-determination. Furthermore, in a performance climate, only the best performers are considered successful [8,16,35]. Consequently, most unsuccessful employees are deprived of opportunities to express themselves or be valued and recognized through their work, and they themselves may not value their work. Additionally, they are constantly compared with their coworkers [37,38] so that outperforming them becomes their most important goal, making it difficult for them to feel a sense of belonging or social support. In summary, in a performance climate, many individuals may suffer from a sense of powerlessness owing to the lack of control, a feeling of meaninglessness, and a sense of isolation owing to the emotional detachment from other people in the organization. Given that they experience powerlessness, meaninglessness, and a sense of isolation despite the huge investment of resources to survive competition and comparisons with others, they may feel alienated and separated from their work. Based on this assumption, this study proposed the following hypothesis:

**Hypothesis** **2a** **(H2a).**
*Performance climate is positively related to work alienation.*


In contrast, a mastery climate values the mastery of skills, focusing on self-development and competence building [16]. Doing so encourages the continuous development of personal competence and enhancement of skills while giving organizational members a sense of control as it focuses on individuals’ right of choice and self-determination [16]. Additionally, a sense of achievement is attained when the current performance level exceeds the previous one. Moreover, using the success criteria based on one’s own achievement [16], it is possible to value one’s work highly without feeling a sense of meaninglessness. This is because individuals can freely express themselves through work by setting and achieving their own goals without the need to compare themselves to their peers [16]. Individuals in a mastery climate consider the comparison and competition among organization members unimportant. Instead, they develop a sense of a shared fate, taking care of their and their coworkers’ interests [8]. Therefore, the sense of isolation can be reduced as they feel socially connected in the process of sharing knowledge with their coworkers before considering their own interests when a person acquires new skills and expertise. Therefore, the alienation of members can be reduced in a mastery climate. Accordingly, this study proposed the following hypothesis:

**Hypothesis** **2b** **(H2b).**
*Mastery climate is negatively related to work alienation.*


### 2.3. The Mediating Effect of Work Alienation

According to the COR theory, individuals try to minimize the net loss of resources when stressed [39]. Those who do not acquire resources by investing in key resources (e.g., time and energy) suffer persistent loss and will face a serious lack of resources [20]. The constant loss of resources may lead to a loss spiral. Resources-limited individuals facing exhaustion may resort to strategies to defend their position by ceasing to invest effort and resources in holding on to the remaining resources [20]. In contrast, without such stress, people are motivated to invest their resources to enrich their repository of knowledge by acquiring new resources [39]. This study suggests that, from the perspective of the COR theory, a performance climate inspires a sense of alienation among the individuals, causing them to implement defensive strategies, which leads to knowledge hiding, while a mastery climate reduces the sense of alienation, making the individuals implement investment strategies to alleviate knowledge-hiding behavior.

Those with a strong sense of alienation are likely to feel separated from what is happening in the work environment and become disillusioned about their work [34], feeling powerless and meaningless, among other things [13,40,41]. In summary, alienation represents a lack of resources [21]. Moreover, a constant feeling of alienation may lead to a chronic lack of resources. This creates resource loss spirals, in which losses gain momentum, creating further losses, making individuals even more vulnerable to losses. Therefore, it is necessary to break these resource loss spirals by protecting the remaining resources. Consequently, individuals suffering from a sense of alienation are likely to hide their knowledge when coworkers ask them to share it to protect themselves from lagging behind their peers competitively. This can be considered a defense strategy to retain their resources as much as possible until they gain new resources or until the situation becomes more favorable in the future.

Conversely, the COR theory suggests that individuals should actively invest their resources for the future when they do not lack current resources [39]. To those who do not feel alienated, hiding knowledge is not the best strategy; by sharing knowledge, these individuals can achieve a higher level of self-improvement by learning and acquiring a new set of skills [5]. Therefore, they will try to develop themselves based on reciprocity by willingly sharing their resources when requested to do so by their coworkers. This can be considered an investment strategy of actively sharing resources with coworkers for self-development and improvement. Accordingly, this study proposed the following hypotheses:

**Hypothesis** **3a** **(H3a).**
*Work alienation mediates the relationship between performance climate and knowledge hiding.*


**Hypothesis** **3b** **(H3b).**
*Work alienation mediates the relationship between mastery climate and knowledge hiding.*


## 3. Methods

To verify these hypotheses, we collected data in more than ten areas of China using the Wenjuanxing system, a Chinese online survey platform. Employees were invited to the current study, and those who agreed to participate were informed that the survey was voluntary and that we would only use this data for research purposes. Data were collected across two different time points. This time-lagged design can help to minimize concerns of common method bias [42]. At time 1, we surveyed 280 individuals regarding motivational climate, work alienation, and demographic information. After excluding those that contained incomplete responses, 245 responses were collected in total (an effective response rate of 87.5%). At time 2 (two weeks later after time 1), the second set of questionnaires asked those who completed responses at time 1 to report their experiences of knowledge hiding. Ultimately, 200 responses, excluding incomplete ones, were used for the analysis (a response rate of 81.6%). Out of the 200 respondents, 114 (57.0%) were female, 113 (56.5%) were younger than 30 years, 175 (87.5%) were full-time employees, and 117 (58.5%) held a four-year bachelor’s degree. The average age of the respondents was 30.47 (SD = 8.27), the average period of service was 5.90 (SD = 6.48). Respondents held various types of jobs (e.g., office administration, sales, technician, R & D, etc.) from a wide range of industries (e.g., manufacturing, finance, construction, information technology, etc.).

### Measures

The questionnaire was translated from the English source text to Chinese and subsequently back-translated into English by two professionals, following the back-translation procedure [43]. All the questions were scored using a seven-point Likert scale (1 = Strongly Disagree, 4 = Neither Agree nor Disagree, 7 = Strongly Agree).

*Performance Climate (Time 1).* To measure the performance climate, this study used eight items of the measuring instrument developed by Nerstad et al. [16] and was widely used in the research on performance climate. The sample items included “In my department/work group, it is important to achieve better than others”, “In my department/work group, internal competition is encouraged to attain the best possible results”, and “In my department/work group, only those employees who achieve the best results/accomplishments are set up as examples”.

*Mastery Climate (Time 1).* A mastery climate was evaluated using the six items of the instrument developed by Nerstad et al. [16] and was widely used, along with the instrument measuring performance climate. The sample items included “In my department/work group, one is encouraged to cooperate and exchange thoughts and ideas mutually”, “In my department/work group, cooperation and mutual exchange of knowledge are encouraged”, and “In my department/work group, each individual’s learning and development is emphasized”.

*Work Alienation (Time 1).* Work alienation was measured using the ten items of the instrument used by Hirschfeld and Feild [44]. The items included “Most of work life is wasted in meaningless activity”, “I feel little need to try my best at work for it makes no difference anyway”, and “I don’t enjoy work; I just put in my time to get paid”.

*Knowledge Hiding (Time 2).* Knowledge hiding was evaluated using the instrument developed by Connelly et al. [6] and was widely used in studies on knowledge hiding. This instrument is divided into the 3 sub-elements—playing dumb, evasive hiding, and rationalized hiding—and each element consists of 4 items, 12 items in all. The sample items included “I agreed to help them but never really intended to”, “I pretended I did not know what they were talking about”, and “I told them that my boss would not let anyone share this knowledge”.

*Control Variables.* Based on previous studies on motivational climate and knowledge hiding, we set age, gender, tenure, education, job position, and job type as control variables.

## 4. Results

Table 1 shows the descriptive statistics and correlation coefficients of the variables used in our study. In addition, to verify the hypotheses, we performed a hierarchical regression analysis and bootstrapping analysis using SPSS 20.0 and the PROCESS macro [45]. First, a regression analysis was performed on Hypotheses 1 and 2. Further, the PROCESS macro analysis was performed 10,000 times to verify the mediating effect of Hypothesis 3 comprehensively. When examining indirect effects, the bootstrapping analysis is widely known to be an effective approach for dealing with distribution imbalances that regression analysis cannot address [45].

H1a proposes that performance climate is positively related to knowledge hiding. To verify H1a, a hierarchical regression analysis was conducted with control variables, performance climate as an independent variable, and knowledge hiding as a dependent variable. Table 2 presents the results of hierarchical regression. The results of the analysis demonstrated that performance climate had a significantly positive (+) effect on knowledge hiding (*β* = 0.48, *p* < 0.001), supporting H1a. H1b predicts that mastery climate is negatively related to knowledge hiding. The results of the analysis demonstrated that mastery climate had a negative (–) effect on knowledge hiding (*β* = −0.51, *p* < 0.001), supporting H1b.

H2a proposes that performance climate is positively related to work alienation. The results of the analysis demonstrated that performance climate had a positive (+) effect on work alienation (*β* = 0.42, *p* < 0.001), supporting H2a. H2b predicts that mastery climate is negatively related to work alienation. The results of the analysis demonstrated that mastery climate had a negative (–) effect on work alienation (*β* = −0.48, *p* < 0.001).

Further, we conducted a Sobel test and bootstrapping methods to assess the indirect effect of work alienation in the relationship between motivational climate and knowledge hiding. Specifically, H3a proposes that work alienation mediates the relationship between performance climate and knowledge hiding. The results of the Sobel test in Table 3 illustrate that the indirect effect of performance climate on knowledge hiding through work alienation is significant (0.31, *p* = 0.00). Moreover, we estimated a 95% bias-corrected confidence interval by bootstrapping 10,000 samples. As shown in Table 3, the confidence interval of knowledge hiding does not include zero (ranging from 0.23 to 0.40), which indicates that the indirect effect was statistically significant. Taken together, H3a is thus supported.

H3b predicts that work alienation mediates the relationship between mastery climate and knowledge hiding. The results of the Sobel test in Table 4 illustrate that the indirect effect of mastery climate on knowledge hiding through work alienation is significant (−0.39, *p* = 0.00). Moreso, as shown in Table 4, the confidence interval of knowledge hiding does not include zero (ranging from −0.49 to −0.30), which indicates that the indirect effect was statistically significant. Taken together, H3b is thus supported.

## 5. Discussion

This study verifies the effect of motivational climate as an antecedent of knowledge hiding and how work alienation mediates this process. The results confirmed that a performance climate positively affects knowledge-hiding behavior, while a mastery climate negatively affects knowledge hiding behavior. The results also confirmed that work alienation plays a partially mediating role between performance climate and knowledge hiding and a partially mediating role between mastery climate and knowledge hiding.

### 5.1. Theoretical and Practical Implications

This study contributes to the existing literature in three respects. First, we identified the role of motivational climate as an antecedent of knowledge hiding. Thus far, research related to knowledge hiding mainly focuses on interpersonal relationships but tends to overlook the crucial role of the organizational environment. Despite its importance, the existing studies on knowledge hiding tend to have treated motivational climate as a moderating effect (e.g., Men et al., Černe et al., Bari et al.) [5,8,46]. Singularly, in his discussion of the moderating effect of motivational climate on the creativity of organization members, Černe et al. [8] indicated the possibility of motivation to work as an antecedent of knowledge hiding by causing mutual distrust between them. This study explored the influence of mastery and performance climates as organizational environments where individuals perform daily tasks. Although studies that have examined the moderating effect of a motivational climate on knowledge hiding (e.g., Men et al., Černe et al., Bari et al.) [5,8,46], the effect of motivational climate was empirically examined as an antecedent of knowledge hiding for the first time in this study.

Second, it is significant that this study integrates two major academic streams: motivational climate and work alienation. In the past, motivational climate and work alienation were studied separately, with little attention paid to work alienation in organizational studies. The contexts of mastery—valuing sharing and cooperation—and performance—emphasizing comparison and competition—can drive organization members in completely different directions in terms of mental status and attitude toward work. In other words, the values emphasized for success in the organizational environment can help ascertain whether individuals will be alienated. Unfortunately, virtually no study has integrated motivational climate and work alienation. Within this context, this study filled this research gap by describing the mechanism of alienation at work behind the relationship between motivational climate and knowledge hiding.

Third, we identified the process through which a motivational climate leads to knowledge hiding, mediated by work alienation based on the COR theoretical framework. Considering that alienation—a concept that embraces powerlessness, meaninglessness, and a sense of social isolation—is represented by a lack of resources [33], the COR theory can help to better understand how work alienation can lead to knowledge hiding [13,21]. Studies have used the COR theory as an important mechanism to explain the stress process in the work environment. According to this theory, individuals try to acquire and maintain resources and protect themselves from loss. If they are persistently exposed to the constant threat of competition and comparison with their coworkers, individuals may become alienated, which is characterized by a state of lack of resources [21]. This may drive them into a loss spiral; resources-limited individuals facing exhaustion may resort to strategies to defend their position by hiding the knowledge resources they have (e.g., Guo et al., Yao et al.) [13,22]. In contrast, in a mastery climate, organization members grow by learning and cooperating with other coworkers, and individuals can use investment strategies to enrich their resources by providing their resources at the request of coworkers based on the norm of reciprocity. Individuals intentionally hide knowledge in a performance climate but are willing to share knowledge in a mastery climate. Thus, this study attempts to examine and understand why individuals hide knowledge by explaining that those alienated because of a performance climate may rely on knowledge hiding by conserving and protecting their remaining resources so they do not fall behind their coworkers. Simultaneously, this study demonstrates that those who do not feel alienated in a mastery climate can implement an investment strategy to share their resources with their coworkers, providing insight into how an organization can reduce knowledge hiding.

This study offers the following implications for organizations and managers. First, given that knowledge hiding is still prevalent, despite the role of knowledge becoming increasingly important in creating and maintaining organizations’ competitive advantage, organizations should create a climate where appropriate criteria for success are presented to set norms that every member can endorse, to reduce knowledge hiding. This can be accomplished by encouraging a mastery climate that emphasizes cooperation, learning, and skill development. For example, organizations need to provide a range of programs through which individuals can develop skills and competence, set a high value on sharing and cooperation, and ensure that success is understood as surpassing oneself rather than others. Organizations can further lower the tendency of hiding knowledge among their members by eliminating a performance climate that encourages comparison and competition.

Second, organizations need to be more concerned about the sense of alienation among the members. This study demonstrates that alienation can mediate the relationship between motivational climate and knowledge hiding. Therefore, it is important to provide organizational support to organization members so that they do not feel powerless, meaningless, or isolated from the organization and their coworkers. Specifically, leaders must engage members in making important decisions, encourage them to set goals for themselves, allow self-expression through work to grant their work a sense of meaning, and promote cooperation and sharing to have a sense of belonging by bonding with their coworkers. Reducing the sense of alienation that the members may feel will have a positive impact at the critical moment when they decide whether to hide or share the knowledge they have.

### 5.2. Limitations and Future Directions

This study’s primary aim is to identify the antecedent of knowledge hiding and its mechanism. Accordingly, this study empirically examines the role of climate, which has not been widely discussed as an antecedent of knowledge hiding and identifies the role of work alienation as a bridge connecting motivational climate and knowledge hiding. It is important to note that our study was able to minimize the issue of reverse causality by designing a time-lagged study conducted at two different points in time. Therefore, the possibility of knowledge hiding influencing alienation or motivational climate can be lowered. However, as in many other previous studies, this study entails limitations that need to be addressed.

The first limitation is the limited number of samples. In this study, we conducted an online survey to broaden the number of research subjects. Therefore, there is a possibility that the samples are limited to a group that can easily access the Internet. However, this concern can be minimized given that most organization members can freely use the Internet regardless of their age. Second, because the survey relies on self-reported data, subjects may have given more socially acceptable responses rather than genuine answers. However, knowledge hiding is concealing knowledge, unlike sharing, so some people might not be able to assess it correctly. Further, self-reporting is the best available method for understanding the phenomenon. Future studies would benefit from introducing diverse methods, including interviews or experimental designs, to enhance the objectivity and depth of the study. Third, there is an issue of the research level. This study intended to identify the mechanism through which the climate of an organization perceived by members influences knowledge hiding among them. Therefore, we used motivational climate as a variable at the individual level. However, as a climate is shared by all the organization members, they are likely to be under a common influence in deciding whether to share or hide knowledge. Therefore, future studies need to verify the effect of motivational climate on knowledge hiding at the group level.

## 6. Conclusions

Despite the efforts an organization puts into encouraging the sharing and dispersion of knowledge among its members, a substantial number of individuals are still unwilling to share their knowledge with other people. Unfortunately, knowledge hiding not only hinders the creation and maintenance of the competitive advantage of an organization but also risks ruining the hider’s interpersonal relationships and impairing performance [6,8,47,48]. Organizations and managers can mitigate knowledge hiding among the members of their organizations by rejecting a performance climate, which encourages comparison and competition while creating a mastery climate that emphasizes learning, skill development, and cooperation.

## Figures and Tables

**Table 1 behavsci-12-00081-t001:** Descriptive Statistics and Correlations.

Variable		Mean	SD	1	2	3	4	5	6	7	8	9	10
1.	Age	30.47	8.27										
2.	Gender	1.57	0.50	−0.26 ***									
3.	Tenure	5.90	6.48	0.85 ***	−0.20 **								
4.	Education	2.68	0.69	−0.53 ***	0.08	−0.42 ***							
5.	Job Position	1.38	0.65	0.58 ***	−0.35 ***	0.58 ***	−0.24 **						
6.	Job Type	2.16	1.34	0.01	0.04	0.03	0.10	0.09					
7.	Performance Climate	4.54	1.11	0.03	−0.15 *	−0.04	−0.07	0.04	−0.09	(0.96)			
8.	Mastery Climate	4.93	1.02	0.03	0.19 **	0.06	0.13	−0.07	0.26 ***	−0.62 ***	(0.95)		
9.	Work Alienation	3.74	0.99	0.01	−0.13	−0.03	−0.15 *	0.02	−0.28 ***	0.73 ***	−0.77 ***	(0.96)	
10.	Knowledge Hiding (T2)	3.59	1.49	0.02	−0.17 *	0.00	−0.15 *	0.01	−0.29 ***	0.79 ***	−0.83 ***	0.93 ***	(0.99)

Note. N = 200. Reliabilities are on the diagonal in parentheses. * *p* < 0.05; ** *p* < 0.01; *** *p* < 0.001 (two-tailed).

**Table 2 behavsci-12-00081-t002:** The Result of Hierarchical Regression.

	Work Alienation		Knowledge Hiding (Time 2)		
	Model 1	Model 2	Model 3	Model 4	Model 5
Step 1: Control Variables					
Age	−0.07	−0.05	−0.13	−0.11	−0.08
Gender	−0.13	0.02	−0.18 *	−0.01	−0.03
Tenure	−0.09	0.06	0.02	0.18 **	0.14
Education	−0.18 *	−0.06	−0.17 *	−0.04	−0.01
Job Position	0.05	−0.03	0.00	−0.09 *	−0.07
Job Type	−0.26 **	−0.11 **	−0.27 ***	−0.11 ***	−0.05
Step 2: Main Effects					
Performance Climate		0.42 ***		0.48 ***	0.25 ***
Mastery Climate		−0.48 ***		−0.51 ***	−0.25 ***
Step 3: Mediator					
Work Alienation					0.55 ***
Overall F	4.35 ***	59.84 ***	4.80 ***	126.33 ***	266.73 ***
R^2^	0.09	0.70	0.10	0.83	0.92
Change in F		199.44 ***		427.34 ***	221.76 ***
Change in R^2^		0.60		0.71	0.09

Note. N = 200. * *p* < 0.05; ** *p* < 0.01; *** *p* < 0.001 (two-tailed).

**Table 3 behavsci-12-00081-t003:** Indirect Effect of Performance Climate on Knowledge Hiding (T2) Through Work Alienation.

	Indirect Effect and Significance Using a Normal Distribution			
Sobel	Effect	SE	z	*p*
	0.31	0.04	7.37	0.00
	Bootstrap results for an indirect effect			
Bootstrap	Effect	Boot SE	LL 95% CI	UL 95% CI
	0.31	0.04	0.23	0.40

Note. N = 200. Bootstrap sample size = 10,000. SE = standard error; LL = lower limit; CI = confidence interval; UL = upper limit.

**Table 4 behavsci-12-00081-t004:** Indirect Effect of Mastery Climate on Knowledge Hiding (T2) Through Work Alienation.

	Indirect Effect and Significance Using a Normal Distribution			
Sobel	Effect	SE	z	*p*
	−0.39	0.05	−7.84	0.00
	Bootstrap results for an indirect effect			
Bootstrap	Effect	Boot SE	LL 95% CI	UL 95% CI
	−0.39	0.05	−0.49	−0.30

Note. N = 200. Bootstrap sample size = 10,000. SE = standard error; LL = lower limit; CI = confidence interval; UL = upper limit.

## Data Availability

Not applicable.

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
