# Peer review of "Effects of Motivational Climate on Knowledge Hiding: The Mediating Role of Work Alienation"

_behavsci, 2022, doi:10.3390/bs12030081_

Round 1

Reviewer 1 Report

research aims empirically discern the relationship between the motivational climate of an organization and the process of hiding knowledge. From this perspective, in my opinion, it is a well-justified paper, well-articulated according to the state of the art, and with an adequate selection of scales to test a set of hypotheses through regression analysis (hypothesis 1 and 2) and bootstrapping analysis (hypothesis 3). From a methodological perspective, it is a well executed investigation analytically.
Now, two aspects I think cause me some concern. First of all, although the different scales were translated using a back-translation process, I think it would be interesting to include the resulting reliability of the items and the scales (alpha of cronbach) . Secondly, I believe that the study suffers from a lack of specificity about the sample (type of organizations, characteristics of the subjects….etcetera). It would be desirable in this regard to include an informative table. In any case, I think it is a interesting study of high interest .

Reviewer 2 Report

Highlighting the subject matter and perspective of the study, knowledge concealment is undoubtedly a key issue in the organisational world. This paper clearly shows how the work climate, mediated by alienation at work, among many other causes, can be an explanatory factor of such behaviour, and clearly justifies its link with the theory of Resource Conservation (RCT).

The most outstanding feature of the article is its design (the theoretical frame of reference it uses is perfectly articulated, very clearly explained and duly related). It is also, as the authors point out, a little-explored approach. The greatest strength of the study is its design, which highlights the relevance of the study and in my opinion justifies its publication. 
With respect to the methodology, being aware that self-reporting is perfectly valid for sample collection, I agree with the authors on the possible limitations of the study, suggesting other alternatives to counteract the results. They may raise doubts about their extrapolation. Nevertheless, they are perfectly justified.
I suggest further specification of the procedure; access and collection of the sample. The explanation "the questionnaire was carried out online in a small company" seems to me to be insufficient. I think that it should be explained how to access the company or companies (not very clear), how to contact the workers, what kind of prior explanation was given to them for their participation in the study, did they know what it was about?  Also better justify the two moments (longitudinal study in terms of the objectives of each period, improve the explanation of why it is done and how these facts have an impact on the results of the study. 
Once these aspects have been improved, I recommend its publication. 
